# Organ-Specific Positron Emission Tomography Scanners for Breast Imaging: Comparison between the Performances of Prior and Novel Models

**DOI:** 10.3390/diagnostics13061079

**Published:** 2023-03-13

**Authors:** Yoko Satoh, Kohei Hanaoka, Chihiro Ikegawa, Masamichi Imai, Shota Watanabe, Daisuke Morimoto-Ishikawa, Hiroshi Onishi, Toshikazu Ito, Yoshifumi Komoike, Kazunari Ishii

**Affiliations:** 1Yamanashi PET Imaging Clinic, Chuo 409-3821, Japan; 2Department of Radiology, University of Yamanashi, Chuo 409-3898, Japan; 3Division of Positron Emission Tomography, Institute of Advanced Clinical Medicine, Kindai University, Osakasayama 589-8511, Japan; 4Kofu Neurosurgical Hospital, Kofu 400-0805, Japan; 5Division of Breast and Endocrine Surgery, Department of Surgery, Kindai University Faculty of Medicine, Osakasayama 589-8511, Japan; 6Department of Radiology, Kindai University Faculty of Medicine, Osakasayama 577-8502, Japan

**Keywords:** organ-specific PET, SiPM-based TOF PET, photomultiplier tube-based PET, dedicated breast PE, phantom study

## Abstract

The performances of photomultiplier tube (PMT)-based dedicated breast positron emission tomography (PET) and silicon photomultiplier tube (SiPM)-based time-of-flight (TOF) PET, which is applicable not only to breast imaging but also to head imaging, were compared using a phantom study. A cylindrical phantom containing four spheres (3–10 mm in diameter) filled with ^18^F-FDG at two signal-to-background ratios (SBRs), 4:1 and 8:1, was scanned. The phantom images, which were reconstructed using three-dimensional list-mode dynamic row-action maximum likelihood algorithm with various β-values and post-smoothing filters, were visually and quantitatively compared. Visual evaluation showed that the 3 mm sphere was more clearly visualized with higher β and smaller post-filters, while the background was noisier; SiPM-based TOF-PET was superior to PMT-based dbPET in sharpness, smoothness, and detectability, although the background was noisier at the SBR of 8:1. Quantitative evaluation revealed that the detection index (DI) and recovery coefficient (CRC) of SiPM-based TOF-PET images were higher than those of PMT-based PET images, despite a higher background coefficient of variation (CV_BG_). The two organ-specific PET systems showed that a 3 mm lesion in the breast could be visualized at the center of the detector, and there was less noise in the SiPM-based TOF-PET image.

## 1. Introduction

^18^F-2-fluorodeoxy-d-glucose (^18^F-FDG) positron emission tomography (PET) has become widely used in the diagnostic imaging of patients with breast cancer (BC), the number of whom has been increasing [1]. As an addition to whole-body PET for the detection of metastases, dedicated breast PET (dbPET) with a photomultiplier tube (PMT)-based detector (Elmammo^TM^; Shimadzu Corp., Kyoto, Japan) was developed, and this system was shown to achieve higher spatial resolution than the whole-body PET/CT by using a highly photon-sensitive detector placed close to the breast and scanning with reduced respiratory motion [2]. dbPET has been demonstrated to be useful in evaluating the effects of neoadjuvant chemotherapy and in screening for BC, especially for the detection of lesions that cannot be detected with other modalities [3,4,5,6].

Another organ-specific PET scanning method has been developed, dedicated head PET (dhPET), which is expected to be useful in the imaging diagnosis of brain diseases, including dementia [7,8].

The newly developed, organ-specific PET device (BresTome™; Shimazu Corp., Kyoto, Japan) is a silicon photomultiplier tube (SiPM)-based time-of-flight (TOF) PET scanner that can switch the position of a cylindrical detector into two imaging positions, one for the head and one for the breast. This means that one scanner can be used for both dhPET and dbPET. Recent studies have evaluated the image quality provided by this scanner as a dhPET system [9,10,11], but regarding its image quality as a dbPET system, we have found no reports of a comparison between this scanner and its prior model, the PMT-based dbPET scanner. We conducted the present phantom study to qualitatively and quantitatively compare the image quality of organ-specific SiPM-based TOF-PET and PMT-based PET for breast imaging.

## 2. Materials and Methods

### 2.1. Organ-Specific, Ring-Shaped PET Scanners

#### 2.1.1. PMT-Based PET Scanner: Elmammo™

The earlier-model dbPET device, a PMT-based PET system named Elmammo™ (Shimadzu, Kyoto), has a transaxial effective field of view (FOV) of 156.5 mm. The scanner has a 195 mm diameter, ring-shaped detector that comprises 36 detector modules arranged in three contiguous rings and the capacity for depth-of-interaction (DOI) measurement [12]. System attenuation correction was calculated using a uniform attenuation map with object boundaries obtained from emission data [13], and scatter correction was performed using the convolution-subtraction method [14] with kernels obtained using background tail fitting. The characteristics and standard performance of this scanner have been reported in detail [2,15].

#### 2.1.2. SiPM-Based TOF-PET Scanner: BresTome™

The new, recently reported model is a SiPM-based TOF-PET system (SET-5002, a prototype of BresTome™; Shimadzu Corp.) with a transaxial 162 mm FOV, which is large enough to scan the brain and breast with sufficient coverage. An innovative feature of this scanner is that the detector can be positioned in a location that is suitable for scanning the whole brain and breast (Figure 1). The scanner characteristics of the two organ-specific PET scanners are summarized in Table 1.

### 2.2. Data Acquisition

We acquired dbPET phantom images using a cylindrical breast phantom with four spheres of various diameters (3, 5, 7.5, and 10 mm; Figure 2). The spheres and the background were filled with ^18^F-FDG solution. The background radioactivity was 4.0 kBq/mL, and the sphere-to-background radioactivity ratios were 4:1 and 8:1. The background radioactivity concentration of the phantom in this study was determined based on the background concentration at 90 min after injection of the clinical dose (3–4 MBq/kg) and the physiological accumulation in the normal mammary gland [2,16].

The breast phantom was positioned so that the spheres were precisely located on the same transverse plane in two different positions (“center” and “periphery”) in the transverse FOV. The two positions in which the spheres were placed were the center and the periphery, i.e., 2 cm below the top edge, of the axial FOV of each detector. This was because although the dbPET image quality has been reported to degrade at <2 cm from the edges of the ring-shaped detector [17], the image quality of both scanners should be comparatively evaluated, since BC near the chest wall is often located in this region.

### 2.3. Image Reconstruction

During data acquisition, both scanners saved both prompt and delayed events in list-mode format, and all images were reconstructed using three-dimensional (3D) list-mode dynamic row-action maximum likelihood algorithm (LM-DRAMA) with one iteration and 128 subsets with image-space point spread function modeling, which is also used in SiPM-based TOF-PET. In LM-DRAMA, relaxation parameter λ depends on the subset number, and the noise propagation from the subset to the reconstructed image is suppressed as the subset number increases, resulting in fast convergence at a reasonable signal-to-noise ratio [18]. Relaxation parameter λ is defined by factor β, which determines the amount of reduction within one iteration. PET images reconstructed with LM-DRAMA, a successive approximation method, are often further smoothed by applying a post-filter, such as a Gaussian filter, for clinical use.

Based on previous findings [19], in the present study, we reconstructed the PMT-based PET images using the β-values of 20 and 100, and we used Gaussian filters of 0, 0.78, 1.17, 1.56, and 1.95 mm full width at half maximum (FWHM) for post-filtering. In contrast, we reconstructed the SiPM-based TOF-PET images using the β-values of 20, 50, 100, and 200, and non-local mean (NLM) post-filters with the smoothing intensities of 0.5, 1.0, and 2.0. Images of both scanners were reconstructed using LM-DRAMA with the β-values of 20, 50, 100, and 200, and NLM post-filters (0.5, 1.0, and 2.0) for SiPM-based TOF-PET images, and with the β-values of 5, 20, 100, 200, and Gaussian post-filters (0, 1.0, and 2.0 mm) for PMT-based PET images.

### 2.4. Analysis of Image Quality

All phantom images were visually evaluated by an experienced nuclear medicine physician (Y.S.) and two experienced PET technologists (K.H. and C.I.) blinded to the reconstruction settings and clinical backgrounds. The transverse images containing the sphere centers were visually evaluated in random order. The images were displayed on an inverse grayscale in a standardized uptake range of 0–4. The images were assessed for the sharpness, smoothness (low noise), and detectability of the 3 mm diameter hot sphere in the phantom using a four-point scale ranging from a score of 0 (not acceptable for diagnosis) to a score of 3 (good). The final score was the mean of the scores given by three readers.

For the quantitative analysis of the phantom images, spherical volumes of interest (VOIs) were placed over the hot spheres of the phantom, with the same size as the inner volume of the spheres. For the calculation of the coefficient of variation of the background (CV_BG_), sixteen 10 mm diameter VOIs were placed in the background region of the slice at the center of the sphere and in slices 1 cm above and below it (for a total of 48 VOIs in three slices). CV is the ratio of the standard deviation to the mean and indicates the degree of variation in uptake intensity relative to the mean. A higher CV_BG_ means that the image has a noisier background with much noise. For the calculation of the detectability index (DI), 28 VOIs (3 mm in diameter) were placed in the background region of each slice (84 VOIs in total). DI and CRC provide information about the visibility and how accurately the system reproduces the true activity concentration, respectively. CV_BG_, DI, and contrast recovery coefficient (CRC) were calculated according to the following equations, respectively:(1)$${\mathrm{CV}}_{\mathrm{BG}}=\frac{{\mathrm{SD}}_{\mathrm{BG},10\mathrm{mm}}}{C_{\mathrm{BG},10\mathrm{mm}}}\times 100\left[\%\right]$$
where SD_BG,10mm_ is the standard deviation of the 48 background VOIs of the 10 mm diameter sphere and C_BG,10mm_ is the average of the 48 background VOIs of the 10 mm diameter sphere.
DI = (C_H,3mm;max_ − C_BG,3mm;mean_)/SD_BG,3mm_(2)
where C_H,3mm;max_ is the maximum standardized uptake value (SUV_max_) of 3 mm diameter VOIs placed on the smallest sphere, C_BG_,_3mm;mean_ is the average of 84 VOIs of 3 mm in diameter placed on the background, and SD_BG,3mm_ is the standard deviation of the 84 background VOIs of 3 mm in diameter.
(3)$$\mathrm{CRC}=\;\frac{\left(C_{H,\;\max}/C_{\mathrm{BG},3\mathrm{mm};\mathrm{mean}}\;\right)-1}{\left(a_{H}/a_{\mathrm{BG}}\right)-1}$$
where C_H,max_ is the SUV_max_ of the VOI of each sphere, and a_H_ and a_BG_ are the activity concentrations in the hot sphere and the background, respectively.

These physical values were calculated as described [19]. The VOI settings are depicted in Figure 3.

### 2.5. Human Imaging

In accordance with the Declaration of Helsinki, human participation in this study was approved by the Institutional Review Board of Kindai University (jRCTs052200055) and Yamanashi PET Imaging Clinic of Kofu Neurosurgery Hospital (no study-specific number; approval date: 3 October 2022).

Patients with breast cancer fasted for ≥6 h prior to the injection of ^18^F-FDG (3 MBq/kg for PMT-based PET) and underwent a 5 min scan of one side of the chest with SiPM-based TOF-PET or PMT-based PET. The scans were performed with the patient lying in the prone position 90 min after ^18^F-FDG injection. The dbPET images were reconstructed based on phantom studies. The SUV_max_ of the patient’s BC lesion(s), the SUV_mean_ of the background mammary gland, and their ratio (tumor-to-background uptake ratio; TBR) were measured. The SUV_mean_ of the background mammary gland was calculated as the average of the SUV_mean_s of six spherical, 10 mm diameter VOIs placed above, below, left of, right of, anteriorly, and posteriorly to the tumor, as close to the tumor as possible (in the cases with lesions at the edge of the FOV near the chest wall, the VOI posterior to the BC was excluded).

All quantitative processes were performed using in-house-modified Metavol software [20].

## 3. Results

### Visual Analysis Results

The phantom images obtained with the two scanners and reconstructed with various β-values and post-filter images are shown in Figure 4. At both 4:1 and 8:1 SBRs, the higher the β-value and the smaller the post-filter were, the noisier the background was. At the 4:1 SBR, the 3 mm diameter sphere was not visible in the images of either scanner, and the 5 mm diameter sphere was also obscured (Figure 4a). When the SBR was 8:1, the 3 mm diameter sphere was visualized, although it was also obscured with lower β-values and larger post-filters (Figure 4b).

In the visual evaluation of the images at the 4:1 SBR (Figure 5a), both scanners showed poor detectability. These results suggest that it is difficult to detect small lesions (~3 mm) with low FDG accumulation. When the SBR was 8:1 (Figure 5b), the SiPM-based TOF-PET device was superior to the PMT-based PET device in sharpness, smoothness (low noise), and detectability. However, at the SBR of 8:1, the SiPM-based TOF-PET images were noisy at the edges of the detector, whereas the PMT-based PET images showed no difference between the center and the edges of the detector. The increased noise in proximal PET thus seemed to have less effect on the detectability of small lesions.

Figure 6 provides the CV_BG_, DI, and CRC results for a phantom with SBRs of 4:1 (Figure 6a) and 8:1 (Figure 6b). CV_BG_ was lower in the central images than in the peripheral images for both scanning systems, and the difference was larger for SiPM-based TOF-PET than for PMT-based PET. In addition, the higher the β-value and the smaller the post-filter were, the higher the CV_BG_ was for the images obtained with either system.

For both scanners, the DI was lower for the peripheral images at the edges than for the images in the center, and in each position for the images with higher β-values. In both the center and the periphery positions of the detector, the DI was also lower for higher β-values. When the 3 mm diameter sphere was sufficiently visible in the images of both systems, at an SBR of 8:1, the DI of the SiPM-based TOF PET device was superior to that of the PMT-based PET device. However, DI was significantly degraded when the SBR was 4:1 for both systems. The DI was higher with larger post-filters in both positions in the PMT-based PET images, and it was highest when the NLM was 1 mm in the SiPM-based TOF-PET images.

The CRCs were higher in the SiPM-based TOF-PET images than in the PMT-based PET images in both positions. However, the 7.5 and 10 mm spheres were overestimated with SiPM-based TOF PET, which was more marked in the peripheral images at an SBR of 4:1.

Representative clinical images of BC patients scanned with SiPM-based TOF-PET and PMT-based PET and reconstructed with different β-values and post-filters are shown in Figure 7. In the clinical images, small breast cancers were clearly visualized using both scanners, regardless of the location of the lesion. Consistent with the results of the phantom study, background noise was increased with higher β-values and smaller post-filters, and these effects on image quality were smaller in SiPM-based TOF-PET. However, the noise at the edge of the FOV on the chest-wall side was more pronounced in SiPM-based TOF-PET.

## 4. Discussion

In this study, SiPM-based TOF PET provided images with less background noise than PMT-based PET images, but both scanners were able to image abnormal ^18^F-FDG accumulations as small as 3 mm. PMT-based PET, however, may have better ability than SiPM-based TOF PET when background accumulation is high and the tumor is located close to the edge of the detector. This is inconsistent with previous studies of whole-body PET that showed that SiPM-based TOF PET provided better sensitivity and spatial resolution than PMT-based PET. This was attributed to PMT-based PET having (i) a smaller-diameter gantry, (ii) smaller crystals and voxel size, and (iii) the ability to perform DOI measurements.

Small breast cancers with high physiological accumulation in the background mammary gland or low FDG accumulation are difficult to identify, even with organ-specific PET, which may be one of the challenges. While the SBR adopted in this study was determined based on previous studies [2], additional studies using phantoms with SBRs of 5:1, 6:1, 7:1, etc., may help determine the clinical application of both scanners to this challenge and define optimal reconstruction conditions for clinical use.

Moreover, referring to the clinical images, the SiPM-based TOF-PET images seemed to have more noise enhanced near the chest wall, especially the ^18^F-FDG abnormal accumulation at the left myocardium (Figure 7). The image quality of ring-shaped dbPET images was reported to be degraded at the detector edge due to FDG accumulated in the myocardium and other body parts, although sensitivity correction was performed [21]. Thus, in SiPM-based TOF-PET with a larger detector surface diameter, radioactivity outside the FOV may have a greater impact. Our present findings thus indicate a trade-off between increasing the detector diameter of SiPM-based TOF-PET—which reduces the risk of missing mammary glands near the chest wall in the FOV—and increasing the noise in this area.

It was reported that the background mammary gland accumulation in PMT-based PET images varies with the menstrual cycle in dense breasts at mammography [16]. However, the nonspecific accumulation of background mammary glands is so low that it does not affect the lesion–background contrast; it is, therefore, not necessary to determine the timing of the PET scan, as is the case with contrast-enhanced MRI [22]. Nevertheless, based on our present observations, it may be necessary to consider the menstrual cycle in SiPM-based TOF-PET scans of patients with dense breasts.

In the dbPET images obtained with either scanning system in the present study, the higher the β-value and the smaller the post-filter were, the higher the CV_BG_ was. This is consistent with another study of PMT-based PET [19]. On the other hand, for the PMT-based PET images in this study, the DI of the 3 mm sphere decreased with smaller post-filters, and this is in contrast to a report indicating that the DI of 5 mm spheres increased with smaller post-filters [19]. This may be because the identification of very small accumulations (~3 mm) is more easily affected by increased CV_BG_. More interestingly, the DI was the highest when the post-filter (NML) was 1 in the SiPM-based TOF-PET images. This suggests that an NML of 1 is the most balanced condition with background accumulation for sphere identification. This is an issue that merits investigations of breast cancer patients to determine the optimal reconstruction conditions for clinical dbPET images.

In the SiPM-based TOF-PET images, SUV_max_ of the 7.5 and 10 mm spheres were higher than the theoretical value, which may be due to Gibbs artifact in the PSF. Although an overestimation of the SUV with PSF correction is known in whole-body PET [23], it is a trade-off between improving the visualization of small lesions and maintaining quantification. It is thus important to understand the characteristics of PSF correction when evaluating clinical breast images obtained with organ-specific PET.

Previous studies of whole-body PET have shown that SiPM-based TOF PET has improved sensitivity and spatial resolution compared with PMT-based PET. Our present CRC results demonstrate that SiPM-based TOF-PET was less affected by the partial volume effect (PVE) than PMT-based PET. The factors that influence the PVE of a PET system are numerous, and the combined result of these factors determines the inherent spatial resolution. Among them, the use of semiconductors in the detector and the improvement of position discrimination with TOF were considered to have significantly contributed to the improvement of spatial resolution by more accurately identifying the crystal position and enabling TOF information to be obtained [24]. As a result, SiPM-based TOF-PET has higher resolution than PMT-PET despite its larger aperture.

Even though PET scanners are more expensive than conventional modalities, whole-body PET scanning using ^18^F-FDG has become widely available because of its broad application to many types of malignant tumors. There is a limit to the targets that can be imaged with organ-specific PET, and the relatively high cost of organ-specific PET systems probably prevents their widespread use. The new organ-specific PET system we evaluated in this study is an innovative PET system that can be used for both dbPET and dhPET, and it is expected to be useful from the viewpoint of medical economics, since the recent increases in breast cancer and dementia are major public health problems; in addition, there is a worldwide need to establish more accurate diagnostic and therapeutic monitoring methods [25,26].

In earlier reports concerning PMT-based PET, one of the issues discussed is the presence of a “blind area,” which is a region close to the chest wall where a part of the mammary gland may be located outside the FOV of the PET. Since our present investigation was mainly a comparison of image quality in phantom studies, the blind area was not examined. SiPM-based TOF-PET has a larger detector surface diameter (28 cm) than PMT-based PET (18 cm). Because of the rounded nature of the human chest wall, the new scanner is expected to have a deeper chest imaging range. Further studies of patients scanned with both PET systems could help determine whether the blind area is sufficiently small. This could contribute to the establishment of a highly accurate breast cancer screening method using dbPET alone with less radiation exposure and less patient distress.

Our study has several limitations. This was primarily an analysis of phantom images and did not include a cohort using patient data. The structure of a patient’s body and that of the phantom significantly differ, and the FDG-accumulated organs (e.g., the left myocardium) were located outside of the FOV in the case of the patient scans. We did not examine the differences between the two scanners regarding the effect of FDG outside of the FOV on the image quality of dbPET. Further studies are needed to determine the optimal clinical reconstruction conditions for the new SiPM-based TOF-PET. In addition, the patients in this study were not scanned with both devices. It may be difficult to conduct a study in which a sufficiently large number of patients are scanned with both systems. An alternative would be to compare subgroups of different patients sorted by age, breast cancer subtype, and other biologic characteristics.

## 5. Conclusions

Both organ-specific PET scanners were shown to have the ability to detect lesions as small as 3 mm in the breast at the center of the detector, although the background of PMT-based PET images was noisier than that of SiPM-based TOF PET images. In contrast, noise at the edges of the FOV was more marked in SiPM-based TOF-PET images, suggesting that small lesions on the chest wall side with low contrast of lesion-to-background uptake might not have been identified. Furthermore, the background noise in the images obtained with both scanners increased with higher β-values and smaller post-filters. These conditions should be determined for each device, as appropriate for clinical use.

## Figures and Tables

**Figure 1 diagnostics-13-01079-f001:**
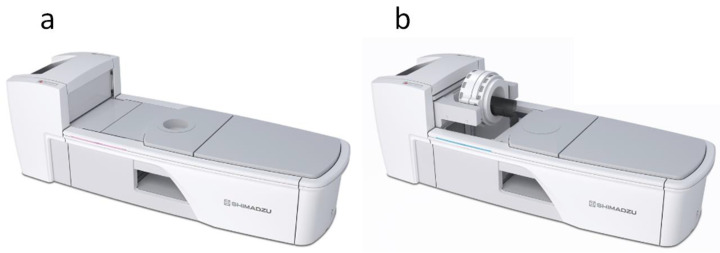
The new, organ-specific, positron emission tomography (PET) system (BresTome™) in the positions for the scanning of the breast (**a**) and the head (**b**).

**Figure 2 diagnostics-13-01079-f002:**
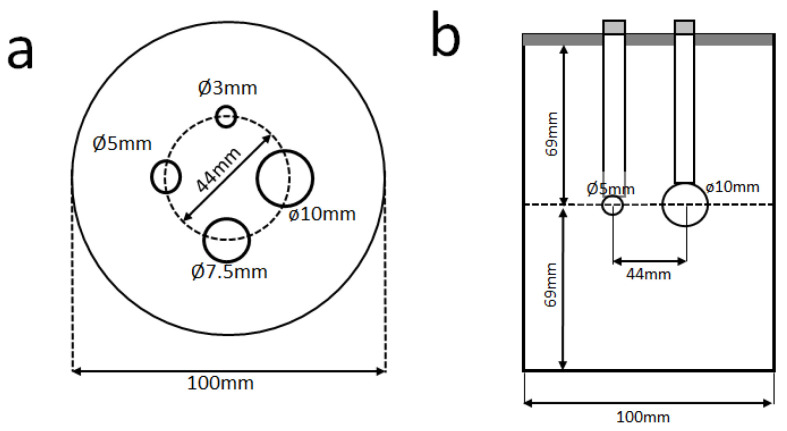
Cross-sections of the phantom with dimensions. Transverse plane view (**a**) and lateral view (**b**) of the breast phantom in which the spheres were arranged. Note: The order of the 7.5 mm diameter and 5 mm diameter spheres is rotated between the two scanners due to an error in the phantom setup.

**Figure 3 diagnostics-13-01079-f003:**
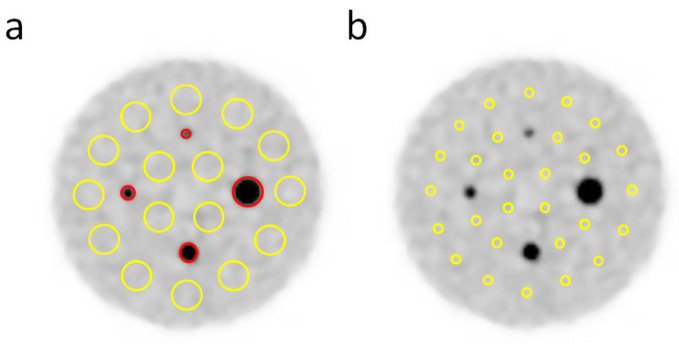
Positioning of the volumes of interest (VOIs) for the analyses of image quality. VOIs measuring 10 mm (**a**) and 3 mm (**b**) in diameter were placed on the phantom background (*yellow*) of the reconstructed images at a sphere-to-background radioactivity ratio of 8:1 and were used for the analysis of the coefficient of variation of the background (CV_BG_), the contrast recovery coefficient (CRC), and the detectability index (DI). In addition, VOIs of the same diameter as each sphere were placed over the hot spheres for the determination of the CRC (a; *red*).

**Figure 4 diagnostics-13-01079-f004:**
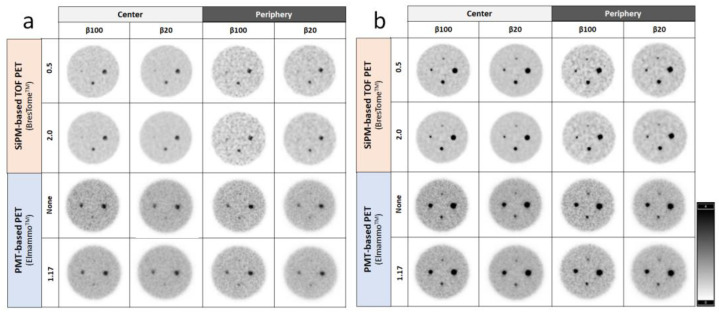
Representative reconstructed images obtained with the two organ-specific PET systems (*top half,* SiPM-based TOF-PET; *bottom half,* PMT-based PET) acquired at the signal-to-background ratios (SBRs) of 4:1 (**a**) and 8:1 (**b**) at the center of the detector (*left half*) and 2 cm below the top edge (*right half*) and reconstructed by applying different β-values (100 and 20) and post-filters of 2.0 and 0.5 as the non-local means (NLM) for SiPM-based TOF-PET (*top row*), and 0 and 1.17 mm as the Gaussian filters for PMT-based PET (*left row*). The smallest of the four spheres (3 mm in diameter) was located at 12 o’clock. Note: The order of the 7.5 mm diameter and 5 mm diameter spheres is rotated between the two scanners due to an error in the phantom setup.

**Figure 5 diagnostics-13-01079-f005:**
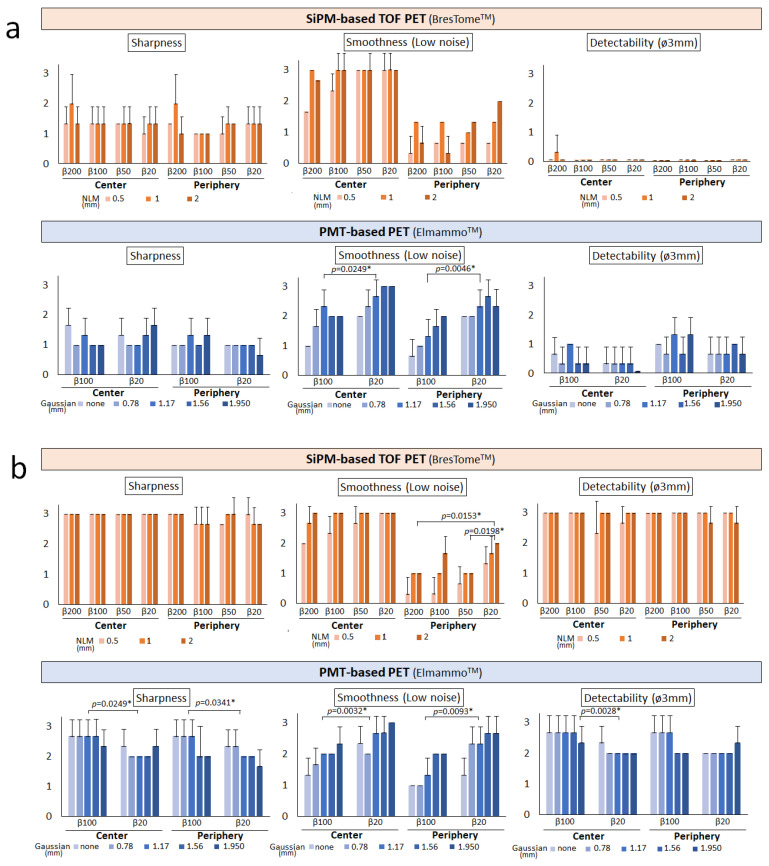
Averaged scores of the visual assessment of breast phantom images obtained with the two organ-specific PET scanners reconstructed with different β-values and post-filters. The sphere-to-background ratios were 4:1 (**a**) and 8:1 (**b**). *Top:* SiPM-based TOF-PET (BresTome™) images. *Bottom:* PMT-based PET (Elmammo™) images. * Statistically significant.

**Figure 6 diagnostics-13-01079-f006:**
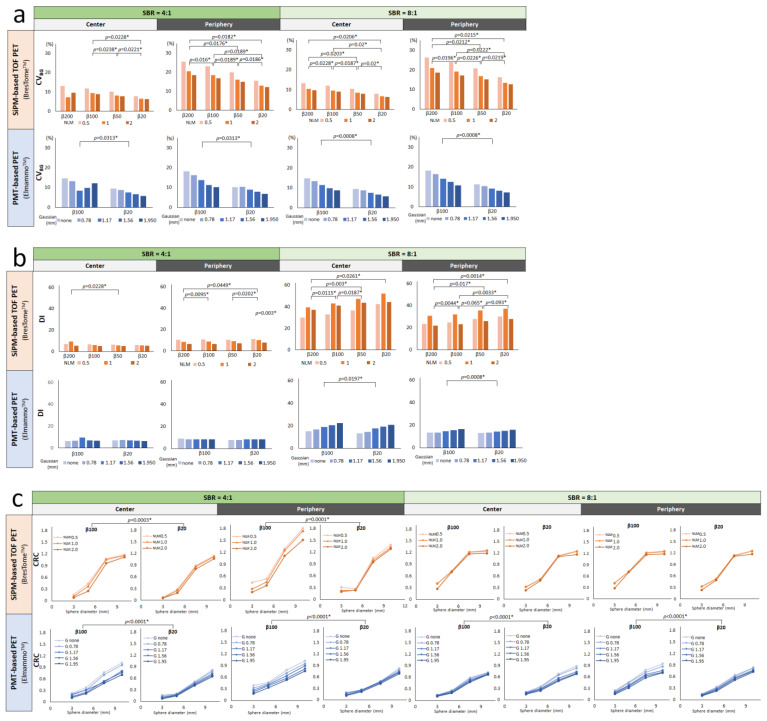
CV_BG_ (**a**), detectability index (**b**), and CRC (**c**) of the breast phantom images with an SBR of 8:1 obtained with the two organ-specific PET scanners. * Statistically significant.

**Figure 7 diagnostics-13-01079-f007:**
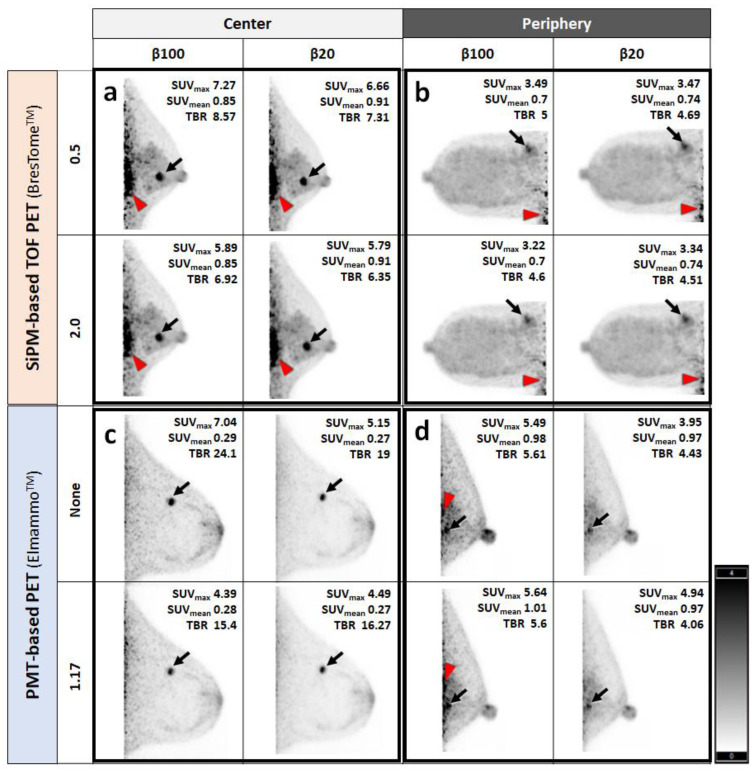
Clinical images of breast cancer (BC) patients scanned with the two organ-specific PET scanners. Medio-lateral maximum intensity projection images of BC near the nipple (**a**,**c**) or near the chest wall (**b**,**d**) scanned with SiPM-based TOF-PET (**a**,**b**) or PMT-based TOF-PET (**c**,**d**). All images presented invasive carcinoma (T1) on an inverted grayscale in a standardized uptake range of 0–4. Small breast cancers were clearly visualized (*arrows*), and noise on the chest wall side was noticeable in some cases (*arrowheads*). (**a**) Left breast of a 56-year-old woman with invasive ductal carcinoma (luminal A). (**b**) Right breast of a 73-year-old woman with invasive ductal carcinoma (HER-2-rich). (**c**) Left breast of a 59-year-old woman with invasive ductal carcinoma (luminal A). (**d**) Left breast of a 48-year-old woman with ductal carcinoma in situ (luminal B).

**Table 1 diagnostics-13-01079-t001:** Scanner characteristics of the silicon photomultiplier tube (SiPM)-based time-of-flight (TOF) scanner and the photomultiplier tube (PMT)-based scanner ^†^.

PET System	SiPM-Based TOF Scanner	PMT-Based Scanner
Product name	BresTome™	Elmammo™
FOV, mm	162	156.5
Transaxial FOV, mm	264	185
Gantry diameter	300	195
Crystal size, mm^3^	2.1 × 2.1 × 15	1.44 × 1.44 × 18
Material of crystal	LGSO	LGSO
Photo device	SiPM	PMT
No. of DOI layers	1	4
TOF technology	Available	n.a.
Matrix size	240 (x) × 240 (y) × 148 (z)	236 (x) × 236 (y) × 200 (z)
Voxel size, mm	1.1	0.78
Spatial resolution at a 5 mm offset in the center of the FOV *	1.0–2.5 mm	≤1.5 mm
Sensitivity at 0 cm in the center of the FOV, cps/kBq *	0.06–0.11 cps/Bq (5–10%)	0.09–0.13 cps/Bq

^†^ Part of the device specifications (including some unlisted information provided by Shimadzu Corp.). * Quantified using NEMA NU 4-2008 ^22^Na source. DOI, depth-of-interaction; FOV, field of view; LGSO, Lu1.8Gd0.2SiO5:Ce; n.a., not applicable; NEMA, National Electrical Manufacturers Association; PET, positron emission tomography; PMT, photomultiplier tube; SiPM, silicon photomultiplier tube; TOF, time-of-flight.

## Data Availability

Not applicable.

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
