# Peer review of "Organ-Specific Positron Emission Tomography Scanners for Breast Imaging: Comparison between the Performances of Prior and Novel Models"

_diagnostics, 2023, doi:10.3390/diagnostics13061079_

Round 1

Reviewer 1 Report

Title:  too long if it can be summarized it is better for readers

Authors Use a cylindrical phantom with four spheres (3-10 mm in diameter) filled with 18F-FDG and two signal-to-background ratios (SBRs) of 4:1 and 8:1, the performance of silicon photomultiplier tube (SiPM)-based time-of-flight (TOF)-PET and photomultiplier tube (PMT)-based dedicated breast PET was compared.

The abstract is clear to me and constructive.

Introduction: clear background presented, and attractive study ration ale exist as well as suitable citation is done. The aim of the study at the introduction section is clear.

Materials and methods: Figures and tables are well presented.

in the section 2-4 regarding analysis of image quality there are some equations that have not cited please cite these questions and number them.

Conclusion: Expand the conclusion to at least list the main findings of your results as possible.

Author Response

Thank you for reviewing our research paper and providing valuable comments. These comments have helped us improve the quality of our manuscript significantly.

Our responses to your comments are given below, in order.

Reviewer 1

  1. Title:  too long if it can be summarized it is better for readers.

→Thank you for this suggestion to simplify the title. In accordance with your comment, we have shortened it to enhance the reader's understanding (page 1, line 2-4), as shown below.

“Organ-specific positron emission tomography scanners for breast imaging: Comparison between the performances of prior and novel models”

  1. Authors Use a cylindrical phantom with four spheres (3-10 mm in diameter) filled with 18F-FDG and two signal-to-background ratios (SBRs) of 4:1 and 8:1, the performance of silicon photomultiplier tube (SiPM)-based time-of-flight (TOF)-PET and photomultiplier tube (PMT)-based dedicated breast PET was compared.

The abstract is clear to me and constructive.

→We are honored to receive such a compliment. Thank you.

  1. Introduction: clear background presented, and attractive study rationale exist as well as suitable citation is done. The aim of the study at the introduction section is clear.

→Thank you for the positive feedback.

  1. Materials and methods: Figures and tables are well presented.

→We are pleased that you appreciate our tables and figure presentations.

  1. in the section 2-4 regarding analysis of image quality there are some equations that have not cited please cite these questions and number them.

→As per your comment, in Materials and Methods:2-4, there was no explanation provided for SDBG,3mm. We have added an explanation for this factor (page5, line 184-185) as follows.

“SDBG,3mm is the standard deviation of the 84 background VOIs of the 3-mm-diameter,”

The explanations for the other parameters included in the three equations given are also shown below each equation. In addition, after careful review, we found errors in the CRC formulas and corrected these as well (page5, line 187).

  1. Conclusion: Expand the conclusion to at least list the main findings of your results as possible.

→We appreciate your suggestion. We have revised the conclusions (page 11, line 397-404), as follows:

Both organ-specific PET scanners were shown to have the ability to detect lesions as small as 3 mm in the breast at the center of the detector, although the background of PMT-based PET images was noisier than that of SiPM-based TOF PET. In contrast, noise at the edges of the FOV was more marked on SiPM-based TOF-PET images, suggesting that small lesions on the chest wall side with low contrast of lesion-to-background uptake might not have been identified. Furthermore, the background noise in the images from both scanners increased with higher β values and smaller post-filters. These conditions should be determined for each device, as appropriate for clinical use.

Reviewer 2 Report

.Averaged scores of the visual assessment of breast phantom images from the 250 two organ-specific PET scanners reconstructed with different β-values and post-filters - How many averages? Is it possible to draw error bars?

Author Response

Dear Reviewer,

Thank you for reviewing our research paper and providing valuable comments. These comments have helped us improve the quality of our manuscript significantly.

Our responses to your comments are given below, in order.

Reviewer 2

Averaged scores of the visual assessment of breast phantom images from the 250 two organ-specific PET scanners reconstructed with different β-values and post-filters - How many averages? Is it possible to draw error bars?

→Thank you for your suggestion. As per your comments, we created a table of means and standard deviations. However, the table was too large, and thus we discontinued the insertion. Accordingly, we have added error bars to Figure 5 (page 7).